# The Application of GPS-Based Friend/Foe Localization and Identification to Enhance Security in Restricted Areas

**DOI:** 10.3390/s24165208

**Published:** 2024-08-12

**Authors:** Lukas Chruszczyk, Damian E. Grzechca, Krzysztof Tokarz

**Affiliations:** Faculty of Automatic Control, Electronics and Computer Science, Silesian University of Technology, Akademicka 16 Street, 44-100 Gliwice, Poland; dgrzechca@polsl.pl (D.E.G.); ktokarz@polsl.pl (K.T.)

**Keywords:** personal localization and identification, airport prone security, GPS positioning, data fusion, surveillance radar

## Abstract

This paper is devoted to the application of object localization and identification with information combined from a radar system and a dedicated portable/mobile electronic device equipped with a global positioning system (GPS) receiver. This device is able to provide object’s (staff member, and staff vehicle) rough location and identification. Such systems are required in very restrictive security areas like airports (e.g., open-air area and apron). Currently, the outdoor area of the airport is typically protected by a surveillance system operated by security guards. Surveillance systems are composed of different sensors, video and infrared cameras, and microwave radars. The sheer number of events generated via the system can lead to fatigue among staff, potentially resulting in the omission of critical events. To address this issue, we propose an electronic system equipped with a wireless module and a GPS module. This approach enables automatic identification of objects through the fusion of data from two independent systems (GPS and radar). The radar system is capable of precisely localizing and tracking objects, while the described system is able to identify registered objects. This paper contains a description of the subsystems of a portable/mobile electronic device. The fusion of information from the proposed system (rough location and identification) with the precise location obtained from short-range radar is intended to reduce the number of false alerts in the surveillance system.

## 1. Introduction

There are many strategic objects that require permanent protection due to their importance, e.g., the airport apron, no man’s land, etc. The national/international border should also be monitored against potential threats. Current security systems include video and infrared cameras, microphones, and, very often, drones, which can reach the target area very quickly and safely without being detected. The information from the sensors is analyzed by qualified personnel, and the decision on further steps is implemented. This approach is very common and effective when the number of events is small. Let us define *an event*: it is a violation of the protected area that *attracts the attention of a security guard* and possibly results in an action. Sometimes an event is trivial, such as an animal walking through the protected area, but it still requires the attention of the person in charge (e.g., tracking the movement on a monitor). An increasing number of alerts can lead to a decrease in the guard’s alertness, increasing the risk of missing the threats. The real problem occurs in the perimeter, where many objects are constantly present and moving. An object can be a car, a passenger walking to the plane, or a member of the airport ground staff. In addition, an airport apron is divided into sub-areas (zones) with different permission levels. Virtual boundaries limit the permitted presence of a particular object. Therefore, the boundary or zone can be easily and unknowingly crossed by a person or object. Therefore, it is necessary to monitor the presence of all objects within the airport and to implement the appropriate procedure in case of an emergency. A sudden intrusion must result in a rapid and appropriate guard response.

Let us focus on the airport’s outdoor security systems, which can be divided into two groups:Internal security: within the terminal area and passenger and baggage services;Outer area security: traffic handling and aircraft runaways (prone), airport infrastructure (fuel tanks and hangars), and perimeter fencing.
The events of 11 September 2001, highlight the contrast between the technical protection inside the terminals and the complete lack of electronic security in the open area of the port—*no perimeter protection*. Perimeter protection is an element of physical security that arises from the possibility of individuals or groups infiltrating critical airport infrastructure. Its importance increases with the terrorist threat to the country concerned. It is assumed that the warranty for such perimeter protection should be 12 years, and the design should take into account advances in the techniques of breaching systems. An airport should be fenced with a minimum height of 2.44 m, signs should be placed every 30 m, a patrol road should be placed on the inside of the fence, and technical underground channels should be built around the fence to allow for installations: lighting, CCTV, and, e.g., motion detection systems. Such a passive system mainly protects against animal intrusion and requires constant monitoring (event validation).

Other examples of perimeter protection elements are the following:Ground (surface) sensors: microwave barriers, infrared sensors, lasers, and radars;Fence-mounted sensors: microphones, vibration detectors, piezoelectric sensors, and fiber optics;Buried sensors: seismic, hydraulic, and magnetic sensors.

The most common fence protection is fiber optic. This allows for continuous signal analysis, precise selection of alarm events, and a wide range of tuning to the environment (e.g., depending on wind strength). Fiber is also resistant to electrical and radio interference. It is also easy to integrate with CCTV systems for remote viewing of the event. Underground sensors, such as the HF400, are placed up to 0.5 m below the ground. Such a system can have two parallel wires (transmitter and receiver) at a certain distance (e.g., 2 m). In this case, the system is invisible (the boundary of the protected area is invisible) and creates a spatial detection zone resistant to accidental damage.

The current and most common strategy is to use vision systems for event analysis with automatic detection, tracking, and recognition of objects within the monitored area. The weakest point of such systems is the requirement of permanent concentration and focusing on many TV displays where a specific event may (or has) occurred. Although the issue has been recognized for many years, there are still no fully satisfactory results applicable to the security field [1,2,3,4,5,6,7,8,9,10]. One of the responsibilities of security personnel is to make a decision whether a detected or observed object has permission to access a particular area of the airport or not. The working conditions are tedious and may lead to the omission of critical events or false alarms. An airport employee can be in a restricted area without being aware of the intrusion—which is not a critical event because of the virtual (i.e., not visually represented) boundaries. The disadvantage of such systems is the lack of automatic differentiation between unauthorized and authorized objects. Despite advances in image and facial recognition algorithms, identification must still be performed manually. Another disadvantage is “serial operation”: only one object can be tracked/identified at a time (by a single camera and operator). In the case of fast surveillance of multiple objects, this requires a multiplication of equipment and operators—and thus costs.

The idea of our system which is presented in the following sections is not limited to the airport but can also be applied in military zones or open areas (e.g., parks), e.g., surrounding hospitals for mentally disabled patients.

This paper is organized as follows:The following subchapters in the introduction section present an overview of civil airport protection requirements and current solutions (Section 1.1), the short-range radar system (Section 1.2), and the used positioning system (Section 1.3);Section 2 presents the idea of data fusion for object localization and identification with an exemplary simulation in (Section GPS Positioning Simulations);Section 3 contains details on the design of the proposed mobile unit (MU) with a radio subsystem design (Section 3.1), GNSS (GPS) subsystem design (Section 3.2), static positioning accuracy measurements (Section 3.2.1), and dynamic positioning accuracy measurements (Section 3.2.2);Section 4 provides details on the design of the proposed communication frame with analysis of the round-trip time in (Section Wireless Communication);Section 5 presents measurements of the power consumption of the mobile unit (MU);The paper concludes with conclusions (Section 6) and acknowledgements.

### 1.1. Civil Airports Protection Requirements and Current Solutions

The European Commission (EC) has created a roadmap of requirements for civil airport protection that will increase the security of critical assets. First, there is the ability to detect threats before they enter the protected zone (outside the airport fence), detection and identification of threats around the protected area, and tracking of objects (unidentified) over the entire surface of the designated area.

The current solution to EC requirements is a microwave radar system that simultaneously detects, tracks, and pinpoints the location of potential intruders. Radar systems also allow continuous monitoring of a wide area, regardless of time and weather conditions. Advanced software solutions analyze the entire perimeter and can track objects beyond the airport fence. The software integrates the radar with cameras, allowing the guard to follow the object on monitors. As mentioned before, this solution is efficient and up to date, but *it leads to routine security behavior*.

The airport has an important characteristic: there are many internal zones with different authorization levels, but literally no protection (physical barrier) can be applied due to the specificity of an airport’s operation: the “flat area” requirement. Sensing techniques can only detect (not protect against) unauthorized access. The best results are achieved by using information from many “complementary” systems [11,12]. A survey of related techniques can be found in [13,14,15]. Table 1 summarizes the advantages and disadvantages of different security solutions for airports.

### 1.2. Radar System

The airport category depends on many aspects. Among various security elements, a short-range radar ensures on-line detection of intruders. However, each radar system (in terms of end-user software application) has functionalities, strengths, and weaknesses. In our example, the wide area protection (airport prone) is implemented by a commercial microwave radar with the following parameters:Operating range up to 1400 m;One second period (a differential “image” where the detection of obstacles is possible).

To improve and enhance the information obtained, radar software can accomplish the following:Reduce reflected signals (echoes);Automatically select objects moving at a speed of at least 0.2 m/s;Integrate the radar (signal) map with the terrain map;Control CCTV (IP), visible and infrared, cameras to provide more accurate automatic motion detection and tracking of selected objects.

The radar system provides 0.25 m (radial) and 1° azimuthal accuracy of a tracked object. These values are “constant” within the radar’s operating range and are therefore significantly better than typical GPS accuracy [15].

The system operator sets the appropriate status of individual objects detected with the system. This is conducted on the basis of the image transmitted via the user interface and other information on the CCTV monitor. Usually, two types of statuses are implemented:Friend—the object is authorized to be present at a certain place and time;Foe—when there is *no certainty* that an object is authorized.

In summary, the location of an object is accurate and automatically found via the radar system, but the status of the object must be set manually, which is *inconvenient and leads to oversights*. In addition, when a large group of objects needs to be classified immediately, the operator must concentrate hard to avoid misclassification. System operators claim that in most cases (over 90%), the system generates false alarms by detecting workers with valid access to a restricted area. As a result, the operator typically reassigns the *friend* status. Such routine scenarios reduce security.

### 1.3. Positioning System

Airport personnel are always equipped with communication systems, such as Tetra transceivers. Each voice or text-based communication device may also have a GPS module installed, which is very useful for locating objects. The advantage of such a system is that employees do not need to carry another device.

Unfortunately, solutions based on the GPS system have disadvantages:GPS module working time. If permanent operation is required, the battery life is limited;Third-party vendors have limited ability to change or update firmware. Communication protocol does not consider system requirements;GPS location accuracy is not sufficient for many objects in close proximity (i.e., a few meters).

The next section presents the airport and industry requirements along with the proposed solution.

## 2. Data Fusion for Object Localization and Identification

Let us start with the real-world requirements of airport security:Location of an object with radar accuracy (resolution);Identification of an object is mandatory;Possible identification of many objects: at least 50 units/second;The personal unit must be portable, with acceptable size and weight;The battery life of the unit must be at least equal to shift time (10 h);The system must be automatic, i.e., the operator must be relieved of the constant need to classify the objects appearing and their status.

The main idea of the presented wireless personal monitoring system arises from the need to locate and identify persons and vehicles in a wide-open outdoor area, e.g., airport apron, military traverse, game field, dock, or pasture.

Among different systems, a fusion of two of them seems to fulfill the provided requirements, i.e., a combination of radar and GPS positioning together with an independent device—an automatic object identifier, which will be called the mobile unit (MU) in short later [16,17]. This system combines the advantages of the location accuracy of a radar system (which, compared to GPS, is significantly better and constant, both temporally and spatially) and possible identification based on the MU equipped with a GPS module (Figure 1). The radar sends the position of an object every one second and must be consistent with a 50 MUs response time (airport requirement). This means that the round-trip time for the position query from the server (PC, e.g., a system operator personal computer) via the base station (BS) to a MU (acquiring the GPS position) and then the round-trip time for the response MU → BS → PC is limited to 20 ms. Such strict timing demands require a special radio front-end and customized transmission protocol design [18].

Thus, the stand-alone system consists of base stations (BSs) and mobile units (MUs) [19,20]. Each MU contains an integrated GPS receiver, an RF transceiver, a power management unit, a microcontroller, and can optionally be equipped with a keyboard, an LCD display, or a biometric sensor (e.g., fingerprint reader) to confirm personal authorization (on demand). The structure of a BS is similar, with the addition of a communication module (e.g., RS-422/485 or Ethernet) to establish a link with the server (PC). A BS has a constant and known location, so it may also contain a GPS receiver to calculate the difference between the real and the reported location. It can be used to correct the location of the MUs (e.g., using simple DGPS techniques). An MU is attached to an object (person or vehicle); therefore, its unique identification number can be assigned to the object. If an MU has a biometric module, the link between the person and the MU can be made automatically (Figure 2).

The idea of automatically assigning object status is derived from a comparison of the locations indicated via radar and the MU (GPS). The proposed algorithm takes into account the movement history and the current difference between the two locations. The most important element is the introduction of two temporary states:*Friend with unknown ID*—for objects (two or more) that are authorized to be in the area. Such a status does not generate an alarm in the system, but all objects will change their status if and only if an ID is assigned. Figure 3 shows such a situation for two objects—if the system’s resolution cannot distinguish between ID1 and ID2, both are tracked without any action. The ID is reassigned if the distance is greater than the system resolution (*d_min_*);*Foe Suspected*—for objects (two or more) that are currently close enough to foe object and none can be uniquely identified. The distance between *friend* and *foe* depends on the accuracy of the system, e.g., radar, GPS, etc. Therefore, it must be customized for a particular system. Figure 4 illustrates the Foe Suspected status that triggers an alarm for the guard.

Obviously, if there are two or more objects with enemy status, the alarm cannot be cancelled and security personnel must check the unknown objects (e.g., remotely via the CCTV system or by sending a patrol in person).

### GPS Positioning Simulations

A simulation of the trajectory of two objects has been performed, which provides some indication of the expected intersection area if the objects can be distinguished (based on GPS positioning accuracy). Assumptions:The objects can be distinguished with 90% statistical probability if the distance between them is at least 2 m;The objects can be distinguished with 50% statistical probability if the distance between them is at least 1.2 m;The speed of both objects is 1 m/s and the GPS positioning rate is 1 per second (1 Hz)—it shows the distance of 1 m between each sample in the direction of the objects’ movement.

Figure 5 shows the trajectories of two objects: *friend* (red) and *foe* (purple). Both objects are initially placed on the left side and move to the right with an assumed speed of 1 m/s. The solid lines mark the boundaries of their positions (for 90% accuracy of the GPS simulator). It can be observed that the region of overlapping positions (indistinguishability region) has, approximately, dimensions of 10 × 10 m (about 100 m^2^ area). This results in a positioning uncertainty of ±5 m, which is the expected value and comparable to measurements with real GPS receivers (presented in the next chapters).

## 3. Design of Mobile Unit (MU)

The requirements placed on the portable, personal device mobile unit (MU), mainly the 20 ms round trip response time, force us to use a communication protocol based on *polling*. The base station (BS) is a master device and initiates the communication (query), while the MU is a slave device and only responds to the query from the BS (MU does not initiate the communication). Such a solution simplifies both MUs and BSs: only one unit can transmit at a time, so there is no need to implement collision detection/collision avoidance (CDCA) techniques. Another important factor is the determinism of the wireless communication time. Mobile units must be small and lightweight: belt or shoulder mounted for personal wear. Other assumptions are listed in Table 2.

### 3.1. Radio Subsystem Design

The wireless communication in the proposed system is based on the license-free 869 MHz band. This band was chosen because, in our case (Europe), it allows 0.5 W (27 dBm) transmit power (for band occupancy below 10% [21,22]). The radio subsystem design starts with a standard link power budget estimation:(1)$$P_{r}=P_{t}+G_{t}+G_{r}-L,$$
where:

*P_r_*—power received via MU [dBm];

*P_t_*—power transmitted via BS [dBm];

*G*_1_—BS antenna gain [dB];

*G*_2_—MU antenna gain [dB];

*L*—communication path loss [dB].

The power loss in the transmission lines between transceiver and antenna (both MU and BS) is neglected because they are directly connected (or by a short cable length). We start with the basic case of free space loss [23,24]:(2)$$L=20\log\left(d\right)+2-\log\left(f\right)+32,$$
where:

*d*—communication distance [km];

*f*—operating frequency [MHz].

Additional correction Δ*L* [dB] is added using the *Weissberger’s foliage model* (i.e., the influence of trees, bushes, etc.) [25]:(3)$$∆L=\left\{\begin{array}{cc} 0.063f^{0.284}d_{f} & 0<d_{f}\leq 14\\ 0.188f^{0.284}d_{f}^{0.588} & 14<d_{f}\leq 400 \end{array}\right.,$$
where *d_f_* [m] is the maximal foliage depth, and *f* [MHz] is the operating frequency.

Alternatively, an *Egli propagation model* [26] takes into account the transmitter and receiver antenna heights:(4)$$L=-20\log\left(\frac{h_{1}h_{2}}{d^{2}}\right)-20\log\left(\frac{40}{f}\right)+122$$
where:

*L*—communication path loss [dB];

*h*_1_—height of the BS antenna above ground [m];

*h*_2_—height of the MU antenna above ground [m].

The calculated values for our cases are shown in Table 3. The cases differ in terms of specific airport location and internal structure (e.g., maximum communication distances), RF modules used, and scenarios (portable MU vs. vehicle-mounted).

The calculated transmitter power *P_t_* and receiver sensitivity values are sufficient for many specific RF modules equipped with on-board power amplifiers. The initial test for case 1 has demonstrated reliable communication at a data rate of 9.6 kbps at a distance of up to 1 km for a radio module 1 (rmod_1, FPX3 RF) and a radio module 2 (rmod_2, ARF-54 RF).

In case 2, a reference design with a radio transceiver and power amplifier (radio module 3, rmod_3) from Texas Instruments was used as a low-cost prototype solution. The data sheet confirms that the required receiver sensitivity should be sufficient for 1.2, 4.8, and 38.4 kbps data rates. The tests performed proved reliable communication at a 1.2 kbps rate and a distance of up to 0.5 km in a medium-density foliage environment (mainly bushes). Open-air measurements extended this range to 1 km.

### 3.2. GNSS (GPS) Subsystem Design

In the presented solution, GNSS positioning is based on GPS satellite navigation. Each navigation device (i.e., GPS receiver) can be characterized by static and dynamic navigation parameters.

The basic static parameter is the position, e.g., expressed in geographic latitude and longitude. The corresponding measurement uncertainty is usually expressed in meters. By analogy, accuracy is the degree of veracity, while precision is related to the degree of reproducibility. This explains why it is more convenient to use statistical analysis: histogram or cumulative histogram.

The basic dynamic navigation parameter is speed. It has been observed that some GPS modules implement various prediction algorithms, which typically reduce the cross-tracking error. Unfortunately, in some cases (e.g., sudden stop combined with weak measurement conditions), this can result in a short non-existing trajectory (over-prediction).

All of the above parameters have a significant influence on the MU design, i.e., a short time-to-fix allows to switch off (power down) a GPS module—which significantly reduces power consumption. Eight GPS modules were tested (all listed in Table 4).

These receivers were selected because of the variety of options for the RF signal source: integrated or external antenna and external active or passive (custom microstripped PCB) antenna. The low price and wide availability were also important factors.

In total, eight measurement cycles were performed, all under practical operating conditions (in the worst case, not more than half of the sky hemisphere was visible). There were no high obstacles or other structures in the immediate vicinity. All measurements were made with a standard frequency of 1 Hz (once per second).

#### 3.2.1. Static Positioning

The static accuracy of selected GPS modules is presented in this section. Portable devices carried by airport personnel (or attached to other objects, e.g., vehicles) usually have an unknown antenna orientation with respect to satellites. For this reason, two test cases were evaluated: GPS antenna pointing up (best case) and down (worst case). The accuracy results led to the selection of the GPS module and its application in the mobile unit. The investigated modules for the chosen antenna selection and its orientation are shown in Figure 6 and Figure 7. All cumulative histograms show a percentage of accuracy with respect to a reference point (with known location).

The static positioning error (calculated for a 10 m circle) for the downward antenna orientation is about 15% worse than for the upward antenna orientation for the module mod_3. For module mod_4, the difference increases to almost 20%. The most interesting result occurs for the mod_2 module: for the upward antenna orientation, 100% of the positions are within the 7 m circle, but for the downward orientation, only 15% are within the 10 m variation (Table 5, Table 6, Table 7, Table 8 and Table 9).

The conclusions are the following: module 2 (mod_2) is superior, but only for upward pointing antennas. Therefore, this module should be considered for use in fixed orientation cases (e.g., vehicle). Modules 3 and 4 (mod_3 and mod_4) should be considered for portable operation.

Cases with an external microstrip antenna (passive, but with a larger area than the built-in internal antenna) and an external active antenna (with approx. 20 dB preamplifier) were also tested. The results (cumulative histograms of positioning error) are shown in Figure 8 and Figure 9.

It can be clearly seen that the active antenna (with preamplifier) improves the positioning accuracy at the cost of higher power consumption (see next section).

The maximum average positioning error occurs for modules mod_5 (built-in antenna down) and mod_1 (microstrip antenna down); its value is more than 50 m, which is an unacceptable result. The best average accuracy is obtained with the mod_3 module (antenna down)—less than 10 m.

Modules mod_3 and mod_4 have the best performance in static measurements. In addition, the mod_4 module is less sensitive to unfavorable antenna orientation.

#### 3.2.2. Dynamic Positioning

A typical scenario is to locate and identify moving objects. The system analyzes all movements and makes a decision about the status of the objects. Therefore, in-motion positioning is much more important than static positioning. Based on our experiments, the positioning accuracy decreases with increasing object speed. Nevertheless, the cross-tracking error of the selected GPS modules was investigated. Measurements were taken with the antenna pointing upwards (i.e., typical operating position). A measurement was taken every 5 s at a distance of 195.5 m at walking speed (i.e., 5 km/h), since most of the tracking objects are expected to be devices carried by airport personnel walking by foot.

Both mod_5 and mod_2 have a tendency to remember and “hold” the last position before power off. This is probably an effect of the internal motion prediction algorithm but is not accessible at the user level.

The mod_1 module works well with the external active antenna. Modules mod_7 and mod_8 have the best performance in static and dynamic (cross-tracking) measurements, outperforming module mod_4. However, their advantage is only visible under clear and open-sky conditions (Figure 10).

In general, the best behavior for static (or low speed) operation (e.g., wearable) is mod_3, mod_4 (both used for the prototypes), and low-power mod_1, but equipped with an additional active antenna. For automotive use (where higher power consumption is not an issue), mod_4, mod_7, and mod_8 receivers are preferred.

#### 3.2.3. Time-to-Fix and Start-Up Times

An additional dynamic parameter indirectly related to navigation accuracy is the time-to-fix, which is the time from power-on (cold start) or power-sleep (hot start) to the first valid position returned. The latency is important and has a great influence on the robustness of the system. It should be noted that *all receivers did not use A-GPS (Assisted GPS) technology in all tests*. Table 10 shows the cold start latency for the selected GPS modules with respect to different antennas, sky conditions, and number of visible satellites (rightmost column). A dash “-” indicates that the GPS module did not report the number of visible satellites.

It can be observed that the common value for cold start is 2–3 min (both cases of built-in antennas). There is a weak relationship between time-to-fix and the number of visible satellites. At the same time, there is a significant relationship between time-to-fix and the strength of the received RF signal (compare cases with internal, microstrip, or active antennae). However, it can be observed that even in the best real-world measurement conditions, the value of time-to-fix (cold start) is worse than indicated in the data sheet (<40 s). This leads to the conclusion that the value of this parameter provided in the manufacturer’s documentation was measured in the best possible working conditions (i.e., open and cloudless sky, no obstacles, and no sources of interference).

The current consumption of all tested modules (default settings) does not exceed 40 mA, while for low-power modules it is about half of this value. However, the low-power modules, such as mod_2, tend to perform worse (due to longer time-to-fix and less precise accuracy).

## 4. Communication Frame

There is a wide range of communication protocols available, all of which have some drawbacks in terms of requirements. The industry partner (airport) pays attention to a constant response time (round trip time = time from query to received response). This results in the polling method and the requirement of a 20 ms round trip time for 50 MU responses per second. This in turn results in the minimum number of bytes in the data frame.

An important part of a mobile device response frame is its ID and GPS coordinates. The simplest form of GPS encoding is a standard NMEA format read directly from the GPS receiver, such as a GGA frame:$GPGGA,100639.000,**5014.8289,N,01850.0039,E**,1,6,1.92,107.8,M,42.2,M,,*5A 
with encoded location **50°14.8289′ N**, **18°50.0039′ E**. Direct retransmission results in the shortest encoding/decoding time and requires at least 19 ASCII characters (bytes), e.g., “**50148289N018500039E**”—not counting start/end of frame/line characters. Note that the encoding is accomplished by the MU’s low-power microcontroller and may increase the response time. However, 19 bytes is still too much for our time-limited communication.

The above frame can be shortened without precision loss using coding with *signed 32-bit integer* representation. The hemisphere (N-S or E-W) is encoded by sign and each pair of ASCII characters is encoded into a single byte. Four bytes with a sign (*int32*) represent a single coordinate, e.g.:50°14.8289′ N → +[50 d] [14 d] [82 d] [89 d] → +[32 h] [0 Eh] [52 h] [59 h] 
where *d* means decimal, *h* is the hexadecimal representation, and sign “+” encodes northern hemisphere (N). The example above assumes NBC (Natural Binary Coding), which is usually not true for real hardware representation, but well supported on the standard compiler level (e.g., C/C++) and simplifies programming. The total number of required bytes to transmit full position (coordinates) is 8. Detailed information about designed custom data frames can be found in [18].

### Wireless Communication

Wireless communication can be solved in different ways, e.g., each source of information, like battery status, GPS position, firmware version, etc., can have its own polling command. This results in short data frames in both directions: BS ↔ MU. Unfortunately, it has one major disadvantage: it creates many wireless data frames and channel occupancies. Note that each payload is encapsulated with a preamble and error detection bits.

The final implementations of the communication protocol [18] have been tested in a realistic operating environment. The results showed that the project constraints were met. The maximum measured query rate (i.e., the next query after receiving the previous response, 100 Mbps Ethernet) was **81 queries per second**. This translates to a minimum total roundtrip time (PC → BS → MU → BS → PC) of **12.3 ms**. The roundtrip time on the “hardware level”, i.e., BS → MU → BS, is **8.5 ms**. Compared to the expected 4.4 ms for wireless communication, 4.1 ms (48%) for the microcontroller units’ processing time is used (Table 11).

A 100 Mbps Ethernet network delay had the least impact on total roundtrip delay: ~0.5–1 ms (4.1–8.1%). It can be noted that high-level software (PC) has introduced ~2.8–3.3 ms delay—over 20% of the total value.

## 5. Power Consumption of Mobile Unit

The first prototype was built with rmod_1 (Radiometrix RF module, Radiometrix Ltd., Harrow, UK) and a standard 2500 mAh Li-Po (lithium polymer) 3.7 V battery (LP356696, size: 3.5 × 66 × 81 mm, Batimex Sp. z o.o., Pruszków, Poland). The power consumption of the prototype submodules is shown in Table 12. The GPS module mod_1 must be supplied with 5 V, so the current consumption is measured at the input of the switching converter 3.3 V → 5 V (~90% efficiency)—Table 12.

The proposed communication protocol and position coding allows 81 requests per second, but the industrial requirement is limited to only 50 requests per second. Therefore, in the worst case, if a single mobile device (out of 50) responds once per second, up to 3% of the time is taken up by transmission and the remaining time (i.e., 97%) is spent listening and waiting for the request. The average power consumption is therefore *I_avg_* = 161 mA. The tested prototype operated for about 12 h, which fulfills the project requirements (at least 10 h).

The second prototype used rmod_2 (Adeunis RF module, Adeunis, Crolles, France) and a smaller 1900 mAh Li-Po (lithium polymer) 3.7 V battery (LP476066, 4.7 mm × 60 mm × 59 mm, Batimex Sp. z o.o., Pruszków, Poland). Table 13 shows the power consumption for the second prototype modules (all supplied at 3.3 V).

Using exactly the same timing assumptions (as for prototype 1), the average current consumption is *I_avg_* = 108 mA. The tested prototype operated for about 14 h, which also meets the project requirements (at least 10 h).

The third prototype of the mobile unit uses rmod_3 (ChipCon family RF modules: CC1101 and CC1190, PitchBook, Seattle, WA, USA) together with a 1400 mAh Li-Po (lithium polymer) 3.7 V battery (LP465555, 4.6 mm × 55 mm × 55 mm, Batimex Sp. z o.o., Pruszków, Poland). Table 14 shows the current consumption of certain modules of the MU (all supplied with 3.3 V).

In this case, the average current consumption is *I_avg_* = 78 mA. The tested prototype operates for approx. 15 h, which again meets the project requirements.

Comparing the three devices (prototypes) presented, the last one has the smallest size and is equipped with a relatively low-capacity battery. Further investigation in the real environment has proven the low power consumption and sufficient accuracy of the device. Moreover, it responds in a reasonable time and the communication range exceeds 1 km, which is sufficient for the early-stage requirements.

## 6. Conclusions

This paper presents an idea for the localization and identification of an object using data fusion. The system has been built, and its Technology Readiness Level is 6 (TRL6), i.e., real-life tests have been performed in a wide-open area.

The system consists of a base station and mobile units with GPS and radio communication modules. A number of research tasks focus us on important requirements but are not listed by the industrial partner. One is to place the mobile unit on the worker’s uniform where it should be attached, i.e., the mobile unit cannot be in the pocket or have free rotation with respect to the sphere due to the GPS antenna. The positioning accuracy depends on the radar system, so the line of sight must be guaranteed, i.e., the positioning accuracy behind obstacles such as cars, airplanes, or even another employee.

The first prototype of the wireless system has acceptable dimensions, and the battery life is much longer than a single work shift. The requirements provided by the industry partner have been met, i.e., low power consumption, sufficiently high communication range, low manufacturing cost, and acceptable physical dimensions. However, the technology is still at Technology Readiness Level 6 (TRL6), and there is still room for improvement in areas such as operating speed and size.

## Figures and Tables

**Figure 1 sensors-24-05208-f001:**
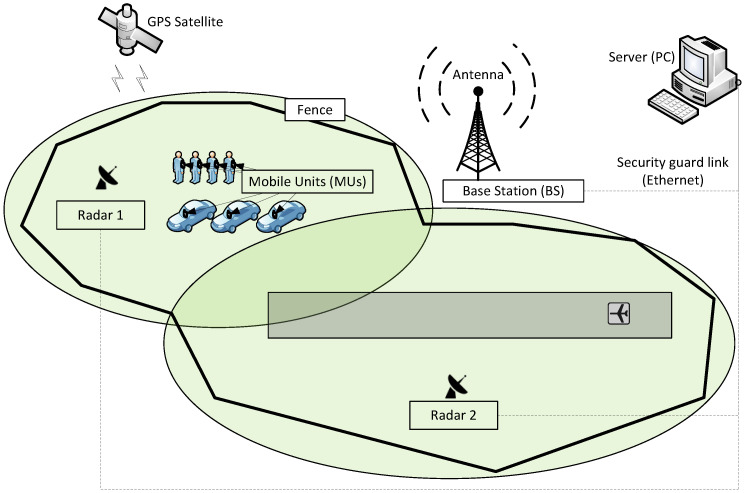
General idea of system for object localization and identification.

**Figure 2 sensors-24-05208-f002:**
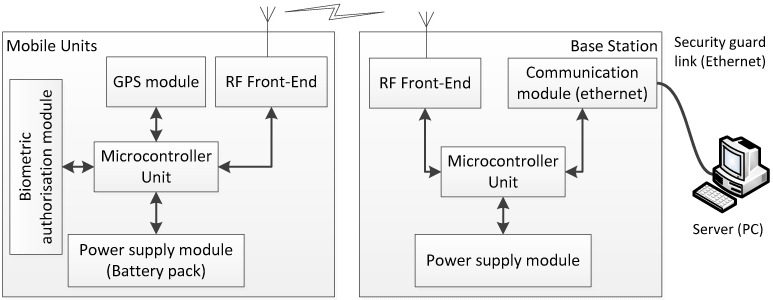
Fundamental modules (structure) for a mobile unit (MU) and a base station (BS).

**Figure 3 sensors-24-05208-f003:**
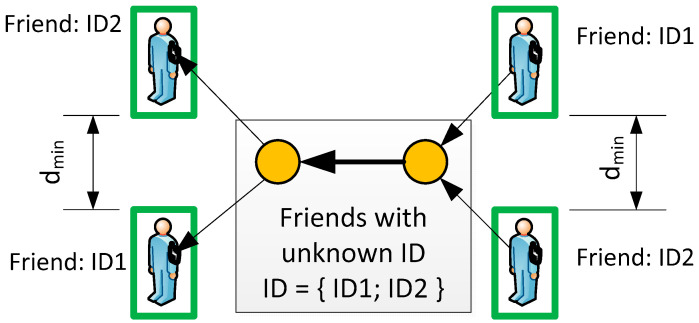
Status of *Friend with unknown ID* (gray box).

**Figure 4 sensors-24-05208-f004:**
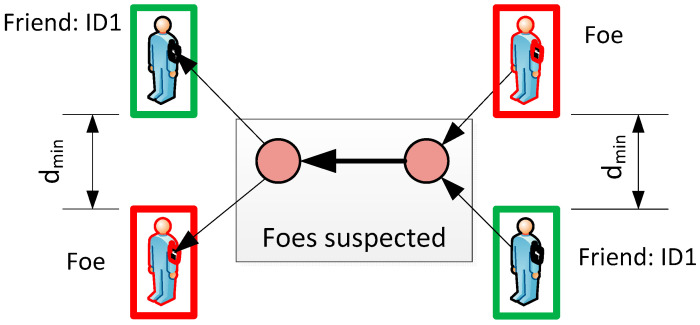
Status examples of *Foe* (red box), *Foe Suspected* (gray box) and *Friend* (green box).

**Figure 5 sensors-24-05208-f005:**
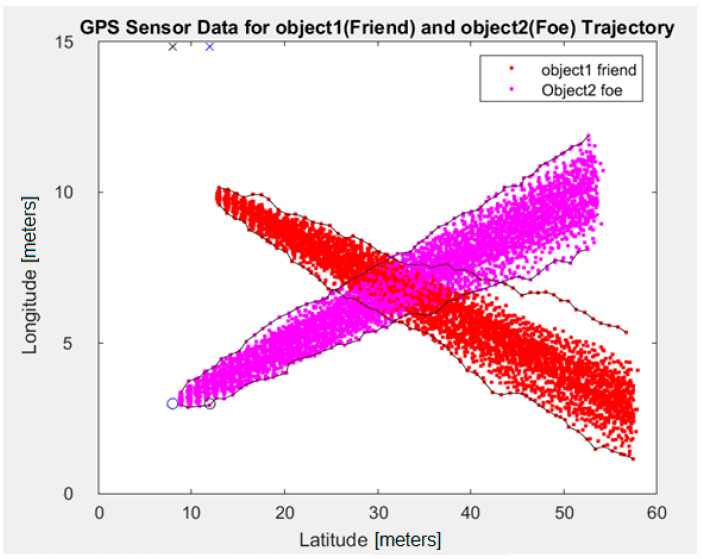
Simulated GPS trajectory with intersection region for two objects.

**Figure 6 sensors-24-05208-f006:**
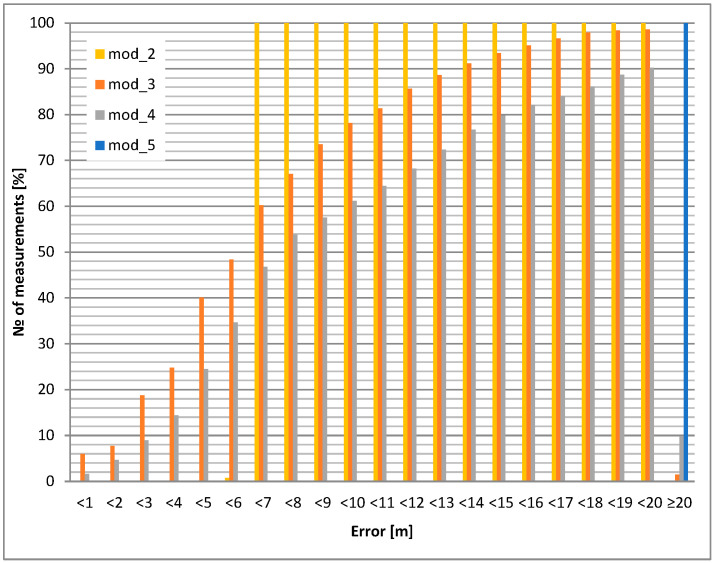
Static positioning error: internal antenna heading upwards.

**Figure 7 sensors-24-05208-f007:**
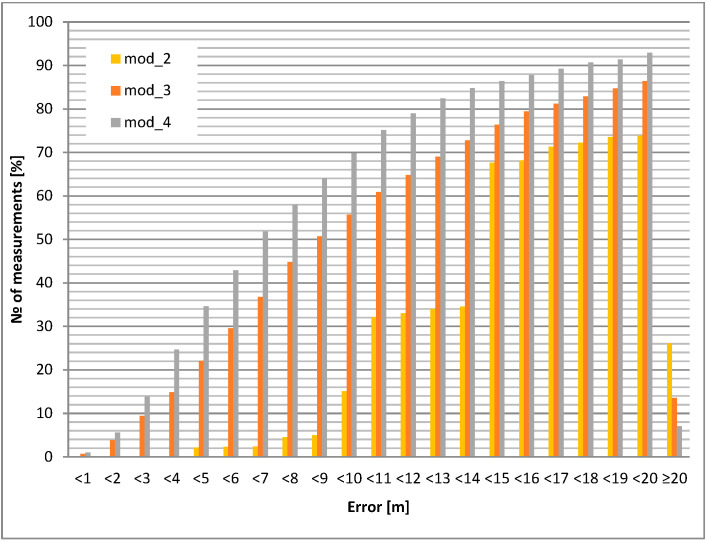
Static positioning error: internal antenna heading downwards.

**Figure 8 sensors-24-05208-f008:**
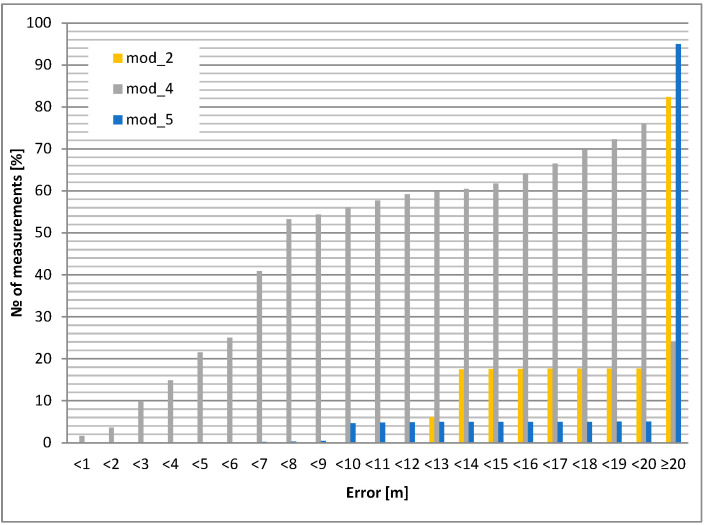
Static positioning error: external microstrip antenna upwards.

**Figure 9 sensors-24-05208-f009:**
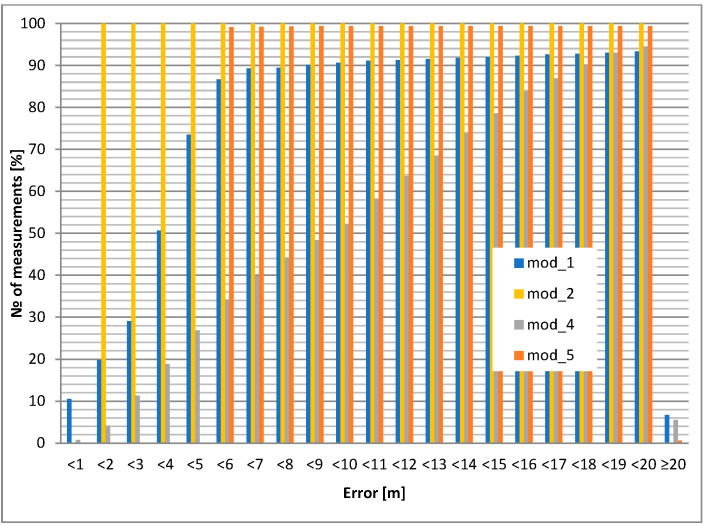
Static positioning error: external active antenna upwards.

**Figure 10 sensors-24-05208-f010:**
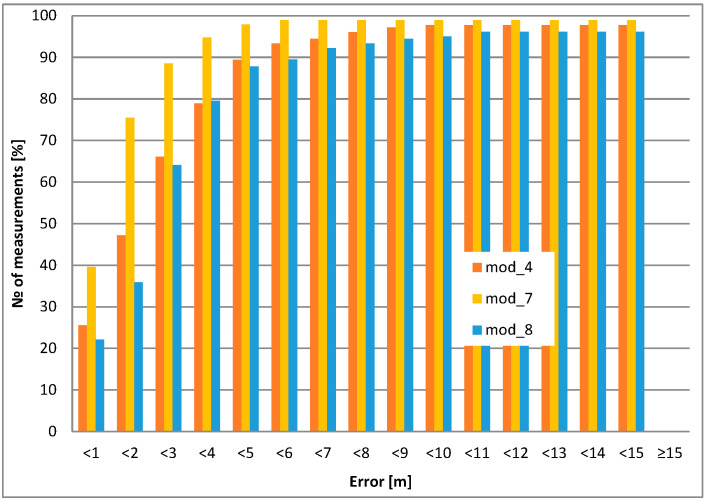
Cross-tracking error.

**Table 1 sensors-24-05208-t001:** Overview of surveillance methods for airport prone.

Method	Key-Points
Passive perimeter protection (fence)	Limited intrusion prevention No intrusion detection
Active perimeter protection (with motion detection systems, e.g., fiber)	Limited intrusion prevention Intrusion detection False alarms possible No intruder tracking after entry
Video surveillance systems, CCTV (visible or infrared)	Intruders/staff location and limited identification Require constant operator attention Single-object tracking (per device) Sensitive to obstacles Visible-light systems are sensitive to environmental conditions (rain or fog)
Video surveillance with automatic motion detection and tracking	Intrusion and staff location and limited identification Operator attendance still required Sensitive to environmental conditions and obstacles
Short-range microwave radars	Intruders and staff accurate automatic location and tracking Sensitive to obstacles (beam shadows and reflections) Direct identification impossible!
Positioning based on satellite navigation, GNSS (e.g., GPS)	Automatic staff location and identification Low positioning accuracy Less sensitive to obstacles Not applicable for intruders!

**Table 2 sensors-24-05208-t002:** Basic system assumptions for all cases.

Assumption	Value	Unit
System capacity (number of MUs)	200	
Maximal positioning time for 50 MUs	1	[s]
Minimal communication distance	2 ^(1)^	[km]
Minimal battery life-time	10	[h]

^(1)^ two cases are considered: case 1—communication distance up to 2 km; case 2—communication distance below 1 km.

**Table 3 sensors-24-05208-t003:** Requirements for a radio subsystem design (for both cases).

Parameter	Case 1 (2 km Distance)	Case 2 (Below 1 km Distance)
Transmitted power [dBm]	26 (0.4 W)	20 (0.1 W)
BS antenna gain [dB]	+5	+7
MU antenna gain [dB]	−1	−2
*d* [km] (2)	2	0.8
*d_f_* [m] (3)	300	400
*ΔL* [dB] (3)	40	44
*h*_1_ [m] (4)	15	18
*h*_2_ [m] (4)	1.5	0.5
*L* [dB] (worst case)	137	133
*P_r_* [dBm] (1) (worst case)	−107	−108

**Table 4 sensors-24-05208-t004:** Selected GPS receivers.

Abbreviation	Chipset
mod_1	MediaTek MT3329 (Pololu Robotics and Electronics, Las Vegas, NV, USA)
mod_2	
mod_3	MediaTek MT3318 (Pololu Robotics and Electronics, Las Vegas, NV, USA)
mod_4	
mod_5	
mod_6	
mod_7	SIRFstarIII (SiRF Commercial Company, San Jose, CA, USA)
mod_8	

**Table 5 sensors-24-05208-t005:** Static location errors for internal antenna heading downwards.

Module		mod_3	mod_4	mod_5	mod_2
Positioning error [m]	min.	0.08	0.08	31.5	5.4
	avg.	9.05	14.0	53.9	8.3
	max.	43.6	74.5	55.4	9.8
№ of measurements		8535	3429	223	2587

**Table 6 sensors-24-05208-t006:** Static location errors for internal antenna heading upwards.

Module		mod_3	mod_4
Positioning error [m]	min.	0.24	0.73
	avg.	14.4	2.86
	max.	93.1	4.92
№ of measurements		4004	580

**Table 7 sensors-24-05208-t007:** Static location errors for external microstrip antenna heading upwards.

Module		mod_4	mod_5	mod_2
Positioning error [m]	min.	0.30	4.03	13.2
	avg.	18.8	26.1	53.0
	max.	67.1	26.7	72.9
№ of measurements		3887	3481	3870

**Table 8 sensors-24-05208-t008:** Static location errors for external microstrip antenna heading downwards.

Module		mod_1
Positioning error [m]	min.	13.2
	avg.	53.0
	max.	72.9
№ of measurements		8535

**Table 9 sensors-24-05208-t009:** Static location errors for external active antenna heading upwards.

Module		mod_1	mod_4	mod_5
Positioning error [m]	min.	0.25	0.09	5.44
	avg.	8.31	11.7	7.13
	max.	55.8	36.1	273
№ of measurements		8535	3489	7115

**Table 10 sensors-24-05208-t010:** Time-to-fix time (cold start).

Module	Antenna	Cold Start [s]	№ of Sats.
mod_5	Internal (half sphere)	194	4
mod_3		131	6
mod_4		378	4
mod_2		195	3
mod_5	Internal (open sky)	90	-
mod_4		40	-
mod_7		75	-
mod_8		70	-
mod_5	Microstrip	691	4
mod_4		226	4
mod_2		1377	3
mod_5	External	92	4
mod_4		357	3
mod_1		35	5

**Table 11 sensors-24-05208-t011:** Influence of selected subsystems on total communication delay.

Delay Source	Delay [ms]	Delay [%]
Wired communication	0.5–1	4.1–8.1
PC processing	2.8–3.3	22.8–26.8
MU + BS processing	4.1	33.3
Wireless communication	4.4	35.8

**Table 12 sensors-24-05208-t012:** Current consumption (in [mA]) for particular modules—prototype 1.

Subsystem/Module	Supply Voltage 5 V	Supply Voltage 3.3 V
Wireless module (transmitting)	500	845
Wireless module (receiving)	50	85
GPS receiver	−	34
Microcontroller	−	19

**Table 13 sensors-24-05208-t013:** Current consumption for particular modules—prototype 2.

Subsystem/Module	Current [mA]
Wireless module (transmitting)	700
Wireless module (receiving)	35
GPS receiver	34
Microcontroller	19

**Table 14 sensors-24-05208-t014:** Current consumption for particular modules—prototype 3.

Subsystem/Module	Current [mA]
Wireless module (transmitting)	138
GPS receiver	34
RF module (receiving)	20
Microcontroller	20

## Data Availability

Data is contained within the article.

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
