# Peer review of "The Application of GPS-Based Friend/Foe Localization and Identification to Enhance Security in Restricted Areas"

_sensors, 2024, doi:10.3390/s24165208_

Round 1

Reviewer 1 Report

Comments and Suggestions for Authors

In this paper, an electronic system for object localization and identification with information combined from a radar system and a dedicated portable/mobile electronic device equipped with a global positioning system (GPS) receiver is proposed to enhance security in restricted area. The modules of the system and the subsystem design are described clearly and logically. This paper is well-organized and technique sound. I think it is ok after completing the following revisions.

1. Some solid lines appear in the paper, such as line 36, line 56, etc., which may need to be removed.

2. I think it would be better to give a chapter arrangement of this paper in Introduction.

Author Response

In this paper, an electronic system for object localization and identification with information combined from a radar system and a dedicated portable/mobile electronic device equipped with a global positioning system (GPS) receiver is proposed to enhance security in restricted area. The modules of the system and the subsystem design are described clearly and logically. This paper is well-organized and technique sound. I think it is ok after completing the following revisions.

Dear Reviewer,

thank you for you review, suggestions and comments.

Comments 1: Some solid lines appear in the paper, such as line 36, line 56, etc., which may need to be removed.

Response 1: The solid lines (text underline) have been replaced by italic characters, where authors believe part of text should be emphasized.

Comments 2: I think it would be better to give a chapter arrangement of this paper in Introduction.

Response 2. The chapter arrangement has been added starting with line 97.

Reviewer 2 Report

Comments and Suggestions for Authors

This manuscript presents a novel object recognition localization method that integrates GPS and radar systems. In the context of airport security, the authors present solutions for currently available recognition systems, including radar. The manuscript contains a few minor issues that can be addressed through revision.

1. The initial two paragraphs of the second section, "Data Fusion for Object Localization and Identification," are more appropriate for an introduction, while the concluding paragraph outlines the central research content of this manuscript.

2. Some physical quantities are not italicized, such as those seen on lines 248 and 249.

3. The literature reviewed in the thesis is somewhat dated and lacks the support of more recent literature from the past three to five years.

Author Response

This manuscript presents a novel object recognition localization method that integrates GPS and radar systems. In the context of airport security, the authors present solutions for currently available recognition systems, including radar. The manuscript contains a few minor issues that can be addressed through revision.

Dear Reviewer,

thank you for you review, suggestions and comments.

Comments 1: The initial two paragraphs of the second section, "Data Fusion for Object Localization and Identification," are more appropriate for an introduction, while the concluding paragraph outlines the central research content of this manuscript.

Response 1: The initial two paragraphs of the section "Data Fusion for Object Localization and Identification" contain real-life requirements provided by the airport security office in order to maintain their tasks: locating and identifying persons and vehicles in a wide open outdoor area without possibility of support by physical barriers. These are specific airport security requirements, while in the Introduction section we have rather presented general requirements and common solutions (with pros and cons).

Comments 2: Some physical quantities are not italicized, such as those seen on lines 248 and 249.

Response 2: The quantities have been italicized (now between lines 280-299 and in the tab. 3).

Comment 3. The literature reviewed in the thesis is somewhat dated and lacks the support of more recent literature from the past three to five years.

Response 3. The bibliography has been modified.

Reviewer 3 Report

Comments and Suggestions for Authors

This paper proposes a system for object localization and identification in restricted areas such as airports using a combination of GPS and radar systems. The mobile electronic device equipped with a GPS receiver provides rough localization and identification, while the radar system provides precise localization. The reviewer's main concerns and queries are as follows:

1. The introduction discusses the need for automatic identification and localization systems, but lacks a clear statement highlighting the specific limitations of existing systems that the proposed approach seeks to address. The motivation for the use of GPS and radar fusion is not fully justified, except that it combines the strengths of both systems. A more detailed analysis of why this combination is necessary is lacking.

2. The authors use formulas to calculate communication path loss, but the derivation of these formulas is not shown or referenced. A brief explanation or reference to the origin of these formulas would strengthen the technical rigor.

3. In practical applications, using better components often leads to increased costs, making hardware complexity a crucial factor in evaluating practicality. It is recommended that the author include a comparison of the complexity of the proposed architecture with currently mainstream methods.

4. Because the paper focuses on real-world testing, there are no simulation results to evaluate. However, the real-world test results could benefit from more detailed analysis, including error bars, statistical significance tests, and comparisons with theoretical predictions.

5. It is recommended that the author improve the expression of some long sentences to make them clearer.

6. These papers provide insights into recent advancements in radar-based systems and motion estimation, which can inspire further improvements to the current work.

[1] Integrated Sensing and Communication with Massive MIMO: A Unified Tensor Approach for Channel and Target Parameter Estimation.

[2] Time-Modulated Arrays in Scanning Mode Using Wideband Signals for Range-Doppler Estimation With Time-Frequency Filtering and Fusion

Comments on the Quality of English Language

The quality of the English language in the document is generally good, with a professional and academic tone. However, there are a few areas where the writing could be improved for clarity and conciseness:

1.     Some terms are used interchangeably, which could lead to confusion. For example, "Mobile Unit" and "MU" are used to refer to the same device. While abbreviations are common in technical writing, defining them when first introduced would help readers unfamiliar with the specific context.

2.     A few sentences could be restructured for better readability. For example, in the introduction, some sentences are long and contain multiple clauses, which makes them difficult to follow. Breaking these sentences into shorter, more focused clauses would improve clarity.

3.     In some places, the language could be more precise. For example, when discussing the advantages and disadvantages of different surveillance methods, the terms "limited" and "possible" are used in a general sense. Providing more specific examples or metrics would strengthen the arguments.

4.     Adding transition words and phrases to connect sentences and paragraphs would make the document flow more smoothly. For example, using phrases like "Furthermore", "Additionally", or "In contrast" would help guide the reader through the logical progression of the ideas.

5.     Ensuring consistency in formatting, such as the use of capitalization, bullet points, and numbering, would enhance the professional appearance of the document.

Overall, with some minor revisions to improve clarity, consistency, and precision, the English language in the document would meet the standards expected of a high-quality academic publication.

Author Response

Dear Reviewer,

thank you for you review, suggestions and comments.

This paper proposes a system for object localization and identification in restricted areas such as airports using a combination of GPS and radar systems. The mobile electronic device equipped with a GPS receiver provides rough localization and identification, while the radar system provides precise localization. The reviewer's main concerns and queries are as follows:

Comments 1: The introduction discusses the need for automatic identification and localization systems, but lacks a clear statement highlighting the specific limitations of existing systems that the proposed approach seeks to address. The motivation for the use of GPS and radar fusion is not fully justified, except that it combines the strengths of both systems. A more detailed analysis of why this combination is necessary is lacking.

Response 1: The basic properties of i.e. microwave short range radar and GPS positioning can be found in the table 1 (Overview of surveillance methods for airport prone) – last two rows. Then, sub-chapters 1.2 (Radar System) present properties of the available radar and sub-chapter 1.3 (Positioning System) present what we can expect from GPS positioning. General conclusions are following:

  • radar can precisely locate objects (0.25 m radial and 1° azimuthal accuracy) and track them (both static and moving) with help of proprietary software (part of complete radar system),
  • radar cannot identify objects: it is unable to distinguish between authorized (“friend”) or non-authorized (“foe”) objects,
  • radar is sensitive for blind zones (shadows),
  • GPS positioning is less accurate (typically a few to several meters) and less sensitive for blind zones,
  • device equipped with GPS receiver can wirelessly transmit own position and identifier, thus “friend” position and identification is possible. A “foe” identification is possible for person (vehicle) equipped with proposed device, but entering zone not being authorized to.

The main idea is to combine accurate positions of unknown objects from radar with known objects with inaccurate positions. And the first step is design and implementation of the custom “GPS based” system for positioning and identification. One of the main problem of such data fusion (congestion of users below GPS positioning accuracy) is presented in the fig. 4.

Comments 2: The authors use formulas to calculate communication path loss, but the derivation of these formulas is not shown or referenced. A brief explanation or reference to the origin of these formulas would strengthen the technical rigor.

Response 2:

  • the (1) is a standard link power budget estimation,
  • the (2) is a basic case of free space loss, referenced by [23,24]
  • the (3) is the Weissberger's foliage model (i.e. the influence of trees, bushes etc.), referenced by [25],
  • the (4) is the Egli propagation model, which takes into account the transmitter and receiver antenna heights, referenced by [26].

Comments 3: In practical applications, using better components often leads to increased costs, making hardware complexity a crucial factor in evaluating practicality. It is recommended that the author include a comparison of the complexity of the proposed architecture with currently mainstream methods.

Response 3: Current mainstream methods for critical civil applications are often based on Tetra system, which is mature, robust, reliable and flexible. However, at the moment of preliminary design, cost of the minimal Tetra infrastructure fulfilling design goals (a single base station and 50 portable/mobile devices transmitting GPS position) were estimated between 70 000 $ and 110 000 $. Majority of the cost (about 80% for our use case) is a single Tetra base station (construction work, hardware, installation and configuration). Therefore there has been decided to develop custom wireless communication system, which has been described in [18-20]. Such approach reduced final implementation cost about tenfold, while still making the system independent from cellular network. The only remaining dependence is a GPS GNSS positioning system, which is the same case for Tetra system (thus positioning accuracy as well). Therefore, presented solution can be received as a cost alternative, while still fulfilling real-life requirements from airport security authorities.

Comments 4: Because the paper focuses on real-world testing, there are no simulation results to evaluate. However, the real-world test results could benefit from more detailed analysis, including error bars, statistical significance tests, and comparisons with theoretical predictions.

Response 4: An additional subchapter 2.1 “GPS Positioning Simulations” been added.

Comments 5: It is recommended that the author improve the expression of some long sentences to make them clearer.

Response 5: We believe the language has been improved, comparing to the previous version.

Comments 6: These papers provide insights into recent advancements in radar-based systems and motion estimation, which can inspire further improvements to the current work.

[1] Integrated Sensing and Communication with Massive MIMO: A Unified Tensor Approach for Channel and Target Parameter Estimation.

[2] Time-Modulated Arrays in Scanning Mode Using Wideband Signals for Range-Doppler Estimation With Time-Frequency Filtering and Fusion

Response 6: Thank you.

Comments on the Quality of English Language

The quality of the English language in the document is generally good, with a professional and academic tone. However, there are a few areas where the writing could be improved for clarity and conciseness:

Comments 1: Some terms are used interchangeably, which could lead to confusion. For example, "Mobile Unit" and "MU" are used to refer to the same device. While abbreviations are common in technical writing, defining them when first introduced would help readers unfamiliar with the specific context.

Response 1: Corrected. The first mention about “Mobile Unit” can be found in the Introduction, together with “MU” abbreviation.

Comments: A few sentences could be restructured for better readability. For example, in the introduction, some sentences are long and contain multiple clauses, which makes them difficult to follow. Breaking these sentences into shorter, more focused clauses would improve clarity.

  1. In some places, the language could be more precise. For example, when discussing the advantages and disadvantages of different surveillance methods, the terms "limited" and "possible" are used in a general sense. Providing more specific examples or metrics would strengthen the arguments.
  2. Adding transition words and phrases to connect sentences and paragraphs would make the document flow more smoothly. For example, using phrases like "Furthermore", "Additionally", or "In contrast" would help guide the reader through the logical progression of the ideas.
  3. Ensuring consistency in formatting, such as the use of capitalization, bullet points, and numbering, would enhance the professional appearance of the document.

Response ad. 2-5: We believe the language has been improved, comparing to the previous version. 

Overall, with some minor revisions to improve clarity, consistency, and precision, the English language in the document would meet the standards expected of a high-quality academic publication.

Thank you.